

# Shifting headlines? Size trends of newsworthy fishes

Fiona T. Francis[1],[*], Brett R. Howard[1],[*], Adrienne E. Berchtold[1],
Trevor A. Branch[2], Laís C.T. Chaves[1], Jillian C. Dunic[1], Brett Favaro[1],
Kyla M. Jeffrey[1], Luis Malpica-Cruz[1], Natalie Maslowski[1],
Jessica A. Schultz[1], Nicola S. Smith[1] and Isabelle M. Côté[1]

[1] Earth to Ocean Research Group, Department of Biological Sciences, Simon Fraser University, Burnaby, BC, Canada
[2] School of Aquatic and Fishery Sciences, University of Washington, Seattle, WA, USA
[*] These authors contributed equally to this work.

Corresponding author
Isabelle M. Côté, imcote@sfu.ca

## ABSTRACT

The shifting baseline syndrome describes a gradual lowering of human cognitive baselines, as each generation accepts a lower standard of resource abundance or size as the new norm. There is strong empirical evidence of declining trends of abundance and body sizes of marine fish species reported from docks and markets.

We asked whether these widespread trends in shrinking marine fish are detectable in popular English-language media, or whether news writers, like many marine stakeholders, are captive to shifting baselines. We collected 266 English-language news articles, printed between 1869 and 2015, which featured headlines that used a superlative adjective, such as 'giant', 'huge', or 'monster', to describe an individual fish caught. We combined the reported sizes of the captured fish with information on maximum species-specific recorded sizes to reconstruct trends of relative size (reported size divided by maximum size) of newsworthy fishes over time. We found some evidence of a shifting baseline syndrome in news media over the last 140 years: overall, the relative length of the largest fish worthy of a headline has declined over time. This pattern held for charismatic fish species (e.g. basking sharks, whale sharks, giant mantas), which are now reported in the media at smaller relative lengths than they were near the turn of the 20th century, and for the largest species under high risk of extinction. In contrast, there was no similar trend for pelagic gamefish and oceanic sharks, or for species under lower risk of extinction. While landing any individual of the large-bodied 'megafish' may be newsworthy in part because of their large size relative to other fish species, the 'megafish' covered in our dataset were small relative to their own species—on average only 56% of the species-specific maximum length. The continued use in the English-language media of superlatives to describe fish that are now a fraction of the maximum size they could reach, or a fraction of the size they used to be, does reflect a shifting baseline for some species. Given that media outlets are a powerful tool for shaping public perception and awareness of environmental issues, there is a real concern that such stories might be interpreted as meaning that superlatively large fish still abound.

## INTRODUCTION

'*The Sea-monsters, mighty of limb and huge, the wonders of the sea, heavy with strength invincible, a terror for the eyes to behold and ever armed with deadly rage—many of these there be that roam the spacious seas . . .*'

Oppian, *Halieutica*

Exploitation continues to have a pervasive impact on fish populations. Globally, fisheries catches peaked in the mid-1990s, and have been relatively static or declining since then (*Pauly & Zeller, 2016*). Remaining abundances of large species are now well below pre-exploitation levels (*Christensen et al., 2003*; *Juan-Jordá et al., 2011*; *Myers et al., 2007*; *Pons et al., 2017*). More locally, declining trends of abundance and/or body sizes of marine fish species are commonly reported from docks and markets (*Richardson et al., 2006*; *McClenachan, 2009*; *To & Sadovy De Mitcheson, 2009*). Smaller fishes in catches and in markets can arise in part from replacement of large, high-trophic level species by smaller ones (*Pauly et al., 1998*) or the targeting of additional small species (*Essington, Beaudreau & Wiedenmann, 2006*), to supplement increasingly less-productive fisheries (*Sethi, Branch & Watson, 2010*). Smaller fish sizes can also result from the targeted removal of the largest individuals within populations through selective fishing (*Richardson et al., 2006*; *McClenachan, 2009*; *Sutter et al., 2012*). Additionally, as fish age and grow they experience more lifetime fishing pressure, resulting in a disproportionate impact of fishing on old, large fish (*Barnett et al., 2017*).

The increasing prevalence of small-bodied fishes in markets or fishing tournaments can influence the public perception of the 'normal' size of fish (*To & Sadovy De Mitcheson, 2009*). This social phenomenon is one type of 'shifting baseline' syndrome. The 'shifting baseline' syndrome describes the general, gradual lowering of human cognitive baselines, as each generation accepts a lower standard of resource abundance as the new norm (*Pauly, 1995*). This has been demonstrated with fish abundance (*Pauly, 1995*); however, there is growing evidence of shifting baselines occurring for other measures of marine resource quality. For example, older fishers often recall both greater fish abundances and larger fish sizes than younger fishers (*Sáenz-Arroyo et al., 2005*; *Ainsworth, Pitcher & Rotinsulu, 2008*; *Lozano-Montes, Pitcher & Haggan, 2008*). They also report more species and fishing grounds as being depleted (*Bunce et al., 2008*; *Katikiro, 2014*). When data are available, these recollections are often confirmed empirically by declining trends in abundance or size (*Richardson et al., 2006*; *Lozano-Montes, Pitcher & Haggan, 2008*).

In this study, we ask whether the widespread trends—real and perceived—in shrinking sizes of marine fish are detectable in popular media, or whether news writers, like many marine stakeholders, have experienced a shifting baseline. Because popular media—television, newspapers, radio, and more recently, the internet—provide sources of information that shape public awareness of environmental issues (*Boykoff, 2009*), a shifting baseline in media reporting of the capture of large fish could bias public perception of the state of marine fish populations. The use of similar superlatives or

hyperbole (e.g. 'giant, 'monster', 'enormous') to describe a very large fish caught some years ago and a relatively smaller fish caught today might be underpinned by a constant journalistic quest for headlines. However, the result among readers might be the perception that very large fish still abound in the ocean.

To address this issue, we surveyed printed English-language news articles about large fish caught. We asked two questions. First, are captured fish described by various size-related superlatives in the English-language printed press truly among the largest specimens reported for each species? Second, is there evidence that the relative size of fish proclaimed to be superlatively large in these news articles declined over time? Such a decline would reflect both the recent rarity of truly large individuals and evidence of 'generational amnesia', that is, the extinction of knowledge over time (*Papworth et al., 2009*), among English-language newsmakers.

## METHODS

### Literature search, selection criteria, and data extraction

To find relevant 'big fish' stories we opportunistically searched online for English-language news articles containing the word 'fish' and any useful superlatives for 'big' (e.g. 'giant', 'huge', 'monster', 'massive', etc.) (Table S1). Other terms, such as 'caught', specific years, or common names (e.g. shark, tuna) were used as needed to narrow down the returned results. We used a variety of online news databases, both public (e.g. Google News) and academic (e.g. Factiva), and news aggregators (e.g. Underwater Times) (Table S1) to retrieve articles. While the databases we searched were not equally efficient at returning usable articles, we typically excluded hundreds of possible articles for each article we retained. For example, a search of the Canadian Newsstand Database on 16 October 2013 returned 5,138 hits of which only 15 were kept in the final analysis. Searches were conducted between February 2013 and December 2015 by most of the authors, with the aim to find articles covering as wide a time span as possible.

Articles were included in our database if they met the following criteria: (i) the article was primarily about a fish that was described as 'big' in some way (Table S1); (ii) the article was found in an English-language news media or digital outdoor magazine (e.g. National Geographic, Field & Stream); (iii) the fish could be identified to species (see in the analysis section for how we dealt with uncertainty surrounding identification of *Mola* spp. and *Mobula* spp.), and (iv) at least one reliable measure of size (length or weight) was given. We considered a measure of size to be reliable if at least one body dimension (length or weight) of the fish was measured, not estimated. Length was assumed to be total length unless otherwise stated. We considered measures of size that were more than twice the maximum verified size of a species listed in either FishBase, an online database of fish life-history information (*Froese & Pauly, 2016*), or by the International Game Fish Association (www.igfa.org) as unreliable and excluded these articles from the dataset. Measures of weight were generally less reliable than measures of length as they were often approximated using other body dimensions (e.g. length × girth) (*Ault & Luo, 2013*). For this reason, we focus primarily on analyses of length data but results pertaining to weight are provided in the Supplemental Information.

When articles met the inclusion criteria, we extracted the location and date of fish capture, the name of the fisher (used to exclude duplicate articles), and the length (cm), weight (kg), or both (if available) of the fish. We used the common name as it was given in the article and any physical descriptions or images to identify the fish to species. On four occasions, we consulted experts to provide or confirm species identification. We excluded cases where species could not be identified with certainty. We recorded whether each fish was caught by recreational (i.e. not for profit) or commercial fishers. In the latter case, we also noted whether the fish was targeted or incidentally caught (i.e. caught as bycatch).

We tried to minimise potential sources of bias and unreliability in the data. Bias was minimised by searching exhaustively, documenting our search (Table S1), and having explicit inclusion criteria such that our search can be replicated by others. An obvious source of bias is our focus on English articles. This was in part for practical reasons because the databases we had access to were in English. It is possible that articles in other languages might present different patterns, although we do not know a priori why this would be the case. One potential source of unreliability is the empirical data extracted from the articles. We reduced unreliability by eliminating records for which species identity was not certain or size was unrealistic, and by using length as main measure of size because it was more likely to be measured than weight. However, it remains possible that sizes were misreported in some of the news article. Such 'mistakes' do not invalidate our analyses because our goal is not to document actual changes in fish sizes but changes in fish sizes reported by journalists. These accounts published in the media are what shapes public perception of the status of fish populations.

It is also important to note that the last decade has seen an exponential increase in both the frequency and accessibility of online media. Because articles originally published in print had to be scanned, uploaded, and searchable for us to detect them, more than half the news items in our database were published since 2000. We can therefore not examine the frequency at which 'big fish' are caught or reported on, since this is confounded with the expansion of digital media.

## Data analysis

We obtained the maximum size (length in cm, and weight in kg) for each species in our database. For length and weight, we used the verified maximum reported for the species in either FishBase or the International Game Fish Association database, or our own records, whichever was largest. Because FishBase often lists more than one length measurement we preferentially used the total length over other measures (Table S2). For lengths we were also able to confirm that species-specific maximum lengths were indicative of truly large fish by comparing them to the average asymptotic length that old fish would approach, $L_{inf}$ (*Froese & Binohlan, 2000*). We obtained $L_{inf}$ from FishBase for all species in our dataset for which this information was available. Maximum lengths and $L_{inf}$ were highly positively correlated ($r = 0.98$, $p < 0.001$, $n = 54$), supporting the assumption that the maximum lengths obtained were representative of true species-specific maximum lengths. We then calculated the relative size for each fish in our database by dividing its reported length (or weight) by the species-specific maximum length (or weight).

We examined the relationships between relative size and the year of publication for all records in our dataset, and then separately, for subsets of species defined on the basis of (1) nominal similarities in taxonomy, habitat use, life history, and/or patterns of exploitation (referred to as 'Species groups') and (2) extinction risk (Table S2). For the first subset, we assigned species, when relevant, to one of three categories. 'Oceanic sharks' included all epipelagic shark species targeted recreationally and/or commercially in different parts of the world. 'Pelagic gamefish' were a similar grouping but for teleost fishes. 'Charismatic megafish' included species that were noted for both their size and relative rarity (e.g. whale sharks, ocean sunfish) and were not usually intentionally targeted by either commercial or recreational fisheries. The disparate species included in this last group (Table S2) made up more than half (55%) of all hits in a Google Image search (15 February 2017) using the terms 'charismatic' AND 'giant' AND 'fish'. The second subset used the current IUCN Red List conservation status of each species (www.iucnredlist.org, 2016) to group species according to their extinction risk. Following *Rodriguez et al. (2015)* we grouped critically endangered, endangered, and vulnerable species together ('High risk'), near threatened and least concern species together ('Low risk'), and data-deficient and not-evaluated species together ('Unknown').

Relationships between relative size and year of publication were tested using quantile regression, since preliminary analysis demonstrated that the variation in relative size was unequal over time (*Koenker & Bassett, 1978*). Quantile regression differs from ordinary least squares regression in that it minimises the sum of absolute values of residual errors around a specified quantile of the dependent variable, rather than the sum of the squares of the residuals around the mean response (*Cade & Noon, 2003*). Exploring a range of quantile responses provides a more complete view of the relationship between the variables of interest. The minimum recommended sample size for quantile regression varies with the quantile selected. Following *Rogers (1992)*, we used a minimum sample size of $n > 5/q$ (where $q$ is the quantile and $q \leq 50\%$; for $q \geq 50\%$, $n > 5/(1 - q)$) and estimated quantile regression functions for the 10th, 15th, 20th, 25th, 50th, 75th, 80th, 85th, and 90th quantiles for the relationships between relative size and year. Analyses were carried out in R, using the QUANTREG package (*R Development Core Team, 2017*).

We calculated the overall average relative length and weight of the fish in our dataset and the average relative lengths and weights of fish in each of our subset groups. To determine if these averages represent truly large fish, we compared them to average relative lengths generated from theoretical fish populations that experience varying levels of fishing pressure as well as sampling effort, two of the factors that can heavily influence the maximum size of fish we would expect to observe (see Supplemental Information 5 for model details).

## Dealing with taxonomic uncertainty of *Mobula* and *Mola*

There was taxonomic uncertainty around species identification in two genera of charismatic megafishes: manta rays, *Mobula* spp., and ocean sunfish, *Mola* spp. We performed sensitivity analyses where possible to examine the potential effect of misindentification on our results.

Eight, possibly nine, species of *Mobula* exist (*Hinojosa-Alvarez et al., 2016*; *White et al., 2018*). Of these, *M. birostris* is the largest (maximum length: 910 cm; *Froese & Pauly, 2016*). Only two of the eight records of *Mobula* in our database included length. One record was larger (640 cm) than the maximum reported size for all but *M. birostris* so was likely correctly identified as *M. birostris*. The other record (457 cm) exceeds the maximum size of all species except *M. birostris*, *M. mobular* (520 cm), and *M. alfredi* (500 cm), but as it was caught in the North Atlantic and outside of the ranges of the latter two species, it was likely also correctly listed as *M. birostris*.

At least three species of *Mola* are known (*Sawai et al., 2017*): *Mola mola* (maximum length: 333 cm), *M. tecta* (242 cm), and *M. alexandrini* (300 cm; *Froese & Pauly, 2016*). There were four records which, based on catch location and size, could have been misidentified as *M. mola*. We therefore re-ran all length analyses using the maximum length of the smallest species, *M. tecta*, instead of *M. mola*. There were no meaningful changes in any of these analyses (see Fig. S1); hence, we present the results assuming that all *Mobula* in the database are *M. birostris* and all *Mola* are *M. mola*.

## RESULTS

We collected 267 news articles that met our criteria and reported either total length and/or weight of a caught fish. These articles spanned more than 100 years, from 1869 to 2015, with over half of the articles ($n = 144$) published between 2000 and 2015. Articles referred to 75 different species of fish (Table S2). Seven of these species had 10 or more news articles; 43 of the species in our dataset were represented by a single news article.

Articles were retrieved from a total of 28 countries, but most came from the USA (45%, $n = 121$), followed by Australia (11%, $n = 29$). While all continents were represented, Africa and South America had the lowest number of articles (five and three articles, respectively). Fourteen of the FAO major fishing areas (MFAs) were represented, ranging from as many as 53 articles (20%) for the Western-Central Atlantic (MFA 31) to as few as a single article for both the Mediterranean and Black Sea (MFA 37) and Southwest Atlantic (MFA 41).

### Trends in relative fish size over time

News articles most commonly reported the total weight of a fish ($n = 238$) rather than total length ($n = 151$). Relative length (i.e. reported length divided by maximum length) of all newsworthy fishes combined decreased over time for the highest relative size quantiles (80th, $p = 0.01$; 90th, $p = 0.02$; 95th, $p = 0.06$), but not for lower quantiles (Fig. 1A; Table S3). In contrast, relative weight did not change significantly over time across all species combined (Fig. 1B; Table S3). There were more relatively small fish reported by weight than by length, with some articles reporting on fish that were less than 10% of the maximum weight for their species (Fig. 1).

When species were divided into subsets of similar species, we found no significant change in relative length for pelagic gamefish (all quantiles, $p > 0.05$; Fig. 2A; Table S4), but a significant increase in the lower quantiles of relative weight across pelagic gamefish species over time (quantiles 10–25%, $p < 0.05$; Fig. S2A; Table S5). For oceanic sharks,
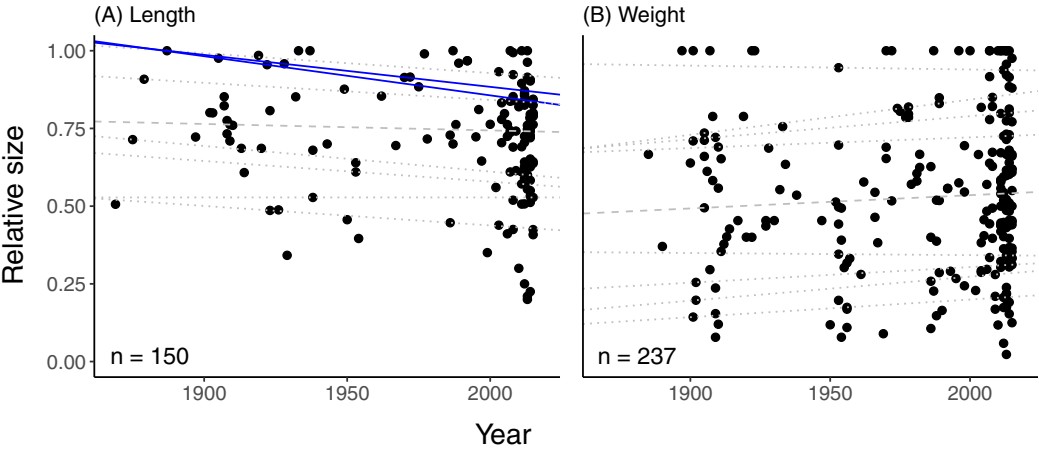

**Figure 1 Relative sizes of fish reported as being exceptional large in printed news headlines.** Sizes of fish reported as being exceptionally large in printed news headlines from 1869 to 2015, relative to the maximum species-specific size. (A) Body length ($n = 150$); (B) total weight ($n = 237$). Lines represent quantile regressions that met a minimum sample size (see Methods). Significant quantile regressions are shown as solid lines; non-significant ($p > 0.05$) quantiles regressions are shown in dotted lines. The dashed line is the 50th quantile.                

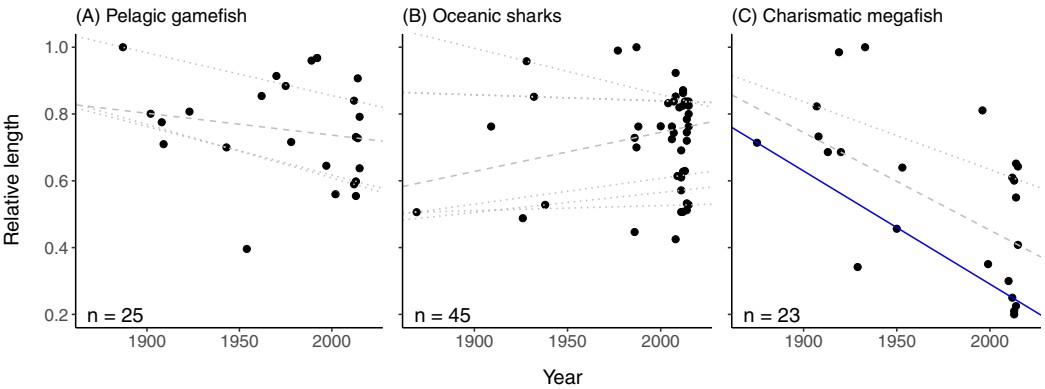

**Figure 2 Relative length of pelagic gamefish, oceanic sharks, and charismatic megafish reported as being exceptionally large in printed news headlines.** Total length of fish reported as being exceptionally large in printed news headlines from 1869 to 2015, relative to the maximum species-specific total length, for three groups of species: (A) pelagic gamefish ($n = 25$), (B) oceanic sharks ($n = 45$), and (C) charismatic megafishes ($n = 23$) The species included in each group are given in Table S2. Lines represent quantile regressions that met a minimum sample size (see Methods). Significant quantile regressions are shown as solid lines; non-significant ($p > 0.05$) quantiles regressions are shown in dotted lines. The dashed line is the 50th quantile.     

there was no change in any of the quantiles in either relative length (Fig. 2B; Table S4) or relative weight (Fig. S2B; Table S5) over time. However, the relative length of charismatic megafish declined significantly over time across six of the nine quantiles examined (i.e. all quantiles except 50th, 75th, and 80th; Table S4), although only the 25th quantile met our sample size ratio threshold (Fig. 2C). The relative weight of charismatic megafishes declined over time only for the 20th quantile ($p = 0.04$; Fig. S2C; Table S5).

There was no clear temporal trend in the relative length (Figs. 3A and 3B; Table S6) or weight (Figs. S3A and S3B) of newsworthy fish at unknown or low risk according to their

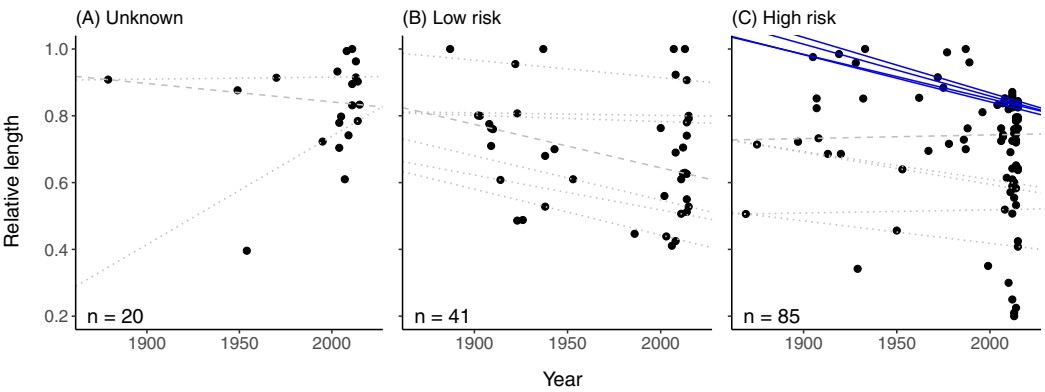

**Figure 3 Total length of fish reported as being exceptionally large in printed news headlines from 1869 to 2015, relative to the maximum species-specific total length, for three categories of extinction risk.** (A) Species of unknown risk ($n = 20$) (i.e. data deficient, not evaluated, least concern); (B) species at low risk ($n = 41$) (i.e. near threatened and vulnerable); (C) species at high risk ($n = 85$) (i.e. endangered and critically endangered). Risk of extinction was derived from the current IUCN Red List. The species included in each group are given in Table S2. Lines represent quantile regressions that met a minimum sample size (see Methods). Significant quantile regressions are shown as solid lines; non-significant ($p > 0.05$) quantiles regressions are shown in dotted lines. The dashed line is the 50th quantile.

IUCN threat status. However, the relative lengths of fishes categorised as being at high risk of extinction declined significantly over time at the higher quantiles (75th–90th quantiles, $p < 0.05$; Fig. 3B; Table S6). There was no detectable change in relative weight of these high-risk species over time (Fig. S3B; Table S7).

## Average relative sizes reported by the media

The average relative length for a species reported by the media was 72.1% ± 18.1% of the maximum species-specific length while the reported average relative weight was only 53.9% ± 25.4% of the maximum species-specific weight.

These averages (mean ± SD) remained similar when data were subset by species groupings (Length: pelagic gamefish = 76.1% ± 15.5%, oceanic sharks = 72.0% ± 15.2%, Weight: pelagic gamefish = 53.1% ± 22.7%, oceanic sharks = 58.2% ± 25.0%) or by IUCN groupings (Length: unknown risk = 82.5% ± 14.4%, low risk = 69.4% ± 17.2%, high risk = 70.3% ± 18.6%; Weight: unknown risk = 46.7% ± 25.4%, low risk = 55.2% ± 24.6%, high risk = 54.0% ± 25.7%). The single exception was charismatic megafish, for which the average relative length in particular was smaller than for other subsets (Length = 56.0% ± 24.0%; Weight: = 40.1% ± 28.4%).

## DISCUSSION

The capture of ginormous fish is often newsworthy. Our results show that fish described as being superlatively large in the printed press are rarely the largest specimens for any species. Fish judged notable by the media were, on average, 72% of the maximum length ever recorded for their species. Moreover, we found some evidence of a shifting baseline syndrome in news media over the last 140 years, as the relative size of the largest fish worthy of a headline has declined over time. This pattern held for large,

charismatic fish species, which are now reported in the media at smaller relative sizes than they were near the turn of the 20th century, and for the largest species under high risk of extinction. In contrast, there was no similar trend for pelagic gamefish and oceanic sharks, or for species under lower risk of extinction.

Is a fish that is 72% of the maximum size reported for its species really very large? The answer depends on the combined effects of life history, fishing pressure, and sampling effort on size-frequency distributions. If we only examine life histories, species that grow quickly, mature early, and die young, will tend to have body size distributions with small tails at large body sizes (*Jennings, Kaiser & Reynolds, 2009*). A sizeable proportion of the population is therefore likely to reach 72% of $L_{inf}$. In contrast, species with slower life histories—those that grow slowly, mature late and die old—will have long-tailed body-size distributions. In such species, the likelihood of reaching 72% of $L_{inf}$ is much smaller, making the individuals that do reach this size remarkably large for their species. However, these size distributions will be heavily influenced by two human-driven factors, fishing pressure and sampling effort, which have opposing effects on the average size of the largest fish, relative to $L_{inf}$, in a population. Older fish experience fishing pressure for more years than younger fish, resulting in disproportionately fewer old fish in populations (*Barnett et al., 2017*), and fishing pressure often also targets large individuals (*Law, 2000*), so one would expect the average largest size of fish to decrease over time. However, higher fishing pressure translates into higher sampling effort, which increases the likelihood of observing a large fish, even in populations where there are few large fish remaining. When we manipulated these two factors in tandem in a simulated fish population, the average size of the largest fish caught was variable across scenarios of fishing pressure and sampling effort (Fig. S4). These results suggest that our observed average relative length of 72% only represents truly large fish under specific conditions of fishing pressures and observational effort. In a population where fishing pressure is high but there are few reported observations, the average relative size of fish will be lower than 72%. Thus, if the fishes in our dataset come from such populations, most fish that make the news are indeed newsworthy. Conversely, if fishes come from populations that are well observed and/or under low fishing pressure, the fishes that make the news at only 72% of maximum size are not particularly large.

We found some evidence of a shifting baseline syndrome among English language news reporters (Fig. 1). Two conditions must be met in order for shifting baseline syndrome to occur. First, biological changes must occur within a system. Second, people's perceptions of a system must be consistent with current biological data, but people must also be unaware that the system was different in the past (*Papworth et al., 2009*). Our study meets the first condition. There is ample evidence of the widespread decline in the abundance and size of large, predatory fishes over the last 50–100 years (*Christensen et al., 2003*; *McClenachan, 2009*; *Myers & Worm, 2003*; *Pauly et al., 1998*; *Sáenz-Arroyo et al., 2005*; *To & Sadovy De Mitcheson, 2009*; *Watson et al., 2012*; *Williams, 2003*). Indeed, this declining trend is shared by species targeted primarily by the commercial fishery (*Myers & Worm, 2003*; *To & Sadovy De Mitcheson, 2009*) as well as by the recreational fishery (*McClenachan, 2009*; *Williams, 2003*), both of which are

represented in our dataset. Our study also partly met the second condition. The relative size of fish proclaimed to be superlatively large in news articles declined over time, but this trend was evident only for the largest newsworthy fishes (80th–90th quantiles for length). This weak declining trend indicates that media perception is at least sometimes consistent with current biological conditions but most of the time, it is not.

There are at least three possible explanations for why we did not find strong evidence for a shifting baseline syndrome in media perception of what constitutes as a newsworthy large fish. First, greater fishing effort (e.g. more vessels) combined with new technologies that allow fishers to travel greater distances (e.g. diesel engines), engage in longer trips (e.g. refrigeration) and find fish more effectively (e.g. echo-locating equipment and helicopters) increases the incidence of fishers encountering truly large fish. Media perception of what constitutes a very large fish might therefore remained unchanged over time because fishers are still able to catch the last few remaining superlatively large individuals by either fishing more intensely (*Anticamara et al., 2011*; *Watson et al., 2012*), by fishing in previously unexploited areas (*Swartz et al., 2010*), by fishing near refuges (e.g. marine protected areas) out of which large fish can stray (*Ahrens, Walters & Christensen, 2012*), or by occasionally catching fish with traits that make them difficult to catch (e.g. shy behaviours; *Biro & Post, 2008*). Second, although the size and abundance of large, predatory fishes have declined over time (*Christensen et al., 2003*; *McClenachan, 2009*; *Myers & Worm, 2003*; *Pauly et al., 1998*; *Sáenz-Arroyo et al., 2005*; *To & Sadovy De Mitcheson, 2009*; *Watson et al., 2012*; *Williams, 2003*), media perception of what constitutes as a superlatively large fish has remained largely unchanged because media perceptions are based primarily on written information sources (e.g. historical records) rather than on the personal experiences of the news writers themselves or on the experiences of fishers, both of which are susceptible to generational amnesia (*Papworth et al., 2009*). Finally, reporting rates of large catches may have vastly increased in recent years with the rise of social media and 24-h online news cycles. The ubiquity of cell phone cameras has increased the public's ability to visually document catches and post them online, so that catches that would not have previously been noticed by journalists are now readily available to be reported on.

Although we cannot fully rule out the first explanation, it is likely that the second had a greater influence on our results. Despite increased fishing effort and technological advances, fish sizes have declined in major commercial markets (*To & Sadovy De Mitcheson, 2009*) as well as in stocks targeted by the recreational fishery (*Williams, 2003*). These observations suggest that changes in fishing effort and technology alone are insufficient to completely mask the recent rarity of truly large fish. The defining principle of shifting baseline syndrome is that humans reset their cognitive baseline of what is considered 'normal' due to a lack of experience about past conditions (*Pauly, 1995*). This lack of past experience may occur because older generations fail to pass on their experience to younger generations (i.e. 'generational amnesia') or because individuals revise their own perceptions of 'normality' based on current conditions while forgetting that they experienced different conditions in the past (i.e. 'personal amnesia'; *Papworth et al., 2009*). The latter explanation for our overall finding is more likely because news

writers rarely rely exclusively on personal experiences or on the experiences of a few individuals to inform decisions about an article. Instead, the practice of 'fact checking' an article against multiple, existing information sources—many of which are in written form such as historical records or databases like FishBase—is common among media professionals (*Elizabeth et al., 2015*). This fact-checking practice ultimately promotes stability in media perception of what is considered a very large specimen for any species, regardless of current conditions. The third explanation may also have influenced our results, but is confounded here by our limited access to pre-digital media.

We surmised that we might detect stronger trends consistent with a shifting baseline syndrome for subgroupings of marine fishes in which the heterogeneity is less pronounced than within our larger, more diverse dataset. However, this was not usually the case. Media opinion of the relative size of fish worthy of a headline remained constant, regardless of all subgroupings used in this study (i.e. pelagic gamefish, oceanic sharks, and species under low or unknown risk of extinction), except for ultra-charismatic fishes and fishes under high risk of extinction. On average, news articles on charismatic species were about individuals only slightly more than half (56%) the maximum possible size for their species. Support for a shifting baseline syndrome in media reporting of this subgroup might be explained, in part, by the fact that many of the species that we classified as being ultra-charismatic are among the largest species of marine fishes (e.g. whale sharks, *Rhincodon typus*; basking sharks, *Cetorhinus maximus*; and giant manta, *Mobula birostris*; Table S2). Landing any individual of these species can be newsworthy simply because of its large size relative to all fishes, rather than relative to its species. These species may also draw additional journalistic interest because of their significance for both conservation and tourism; four of the six megafish in our dataset are threatened with extinction (Table S2), which drove the trend of declining relative length for high-risk species. For journalists, stories about these massive fish may attract a broader audience than the more niche interests of sportfishers. At the same time, declines in average size and abundance have been observed for at least some species within our ultra-charismatic subgroup (*Bradshaw, Mollet & Meekan, 2007*; *Bradshaw et al., 2008*; *Holmberg, Norman & Arzoumanian, 2009*; *Dulvy et al., 2008*). The declining relative size of ultra-charismatic fishes in media reporting therefore reflects current conditions in marine ecosystems. News reporters, however, are less likely to recognize that current ultra-charismatic fishes are smaller than those in the past because for these species, reporters are making size comparisons among all fish species instead of within-species.

## CONCLUSION

Real biological changes have occurred in the world's oceans: the maximum size of many marine fish species has fallen and is expected to continue to decrease (*McClenachan, 2009*; *Cheung et al., 2013*), and large fishes are now generally rarer than in previous decades and centuries (*Christensen et al., 2003*; *Myers & Worm, 2003*). However, the continued use of superlatives to describe fish that are a fraction of the maximum size they could reach, or a fraction of the size they used to be, does not reflect these biological changes, except perhaps for ultra-charismatic fishes. For this group, declines in the

reported size reflect real shifts in body size. Given that media outlets are a powerful tool for shaping public perception and awareness of environmental issues (*Anderson, 2013*), there is a real concern that such stories might be interpreted as meaning that superlatively large fish still abound. This false ocean optimism might in turn diminish support for stricter fisheries management and affect support for marine conservation efforts.

## ACKNOWLEDGEMENTS

The opening quote was borrowed from a paper on sea monsters in Antiquity by Alexander Jaffe (Berkeley Undergraduate Journal of Classics, 2013). We thanks Craig McClain, Alistair Dove and one anonymous reviewer for helpful comments on the MS.

### Funding
Funding provided by a Natural Sciences and Engineering Research Council of Canada (NSERC) Discovery Grant to Isabelle M Côté. The funders had no role in study design, data collection and analysis, decision to publish, or preparation of the manuscript.

### Grant Disclosure
The following grant information was disclosed by the authors:
Natural Sciences and Engineering Research Council of Canada (NSERC) Discovery Grant.

### Competing Interests
The authors declare that they have no competing interests.

### Author Contributions
- Fiona T. Francis conceived and designed the experiments, performed the experiments, analysed the data, prepared figures and/or tables, authored or reviewed drafts of the paper, approved the final draft.
- Brett R. Howard conceived and designed the experiments, performed the experiments, analysed the data, prepared figures and/or tables, authored or reviewed drafts of the paper, approved the final draft.
- Adrienne E. Berchtold performed the experiments, authored or reviewed drafts of the paper, approved the final draft.
- Trevor A. Branch conceived and designed the experiments, analysed the data, prepared figures and/or tables, authored or reviewed drafts of the paper, approved the final draft.
- Laís C.T. Chaves performed the experiments, approved the final draft.
- Jillian C. Dunic analysed the data, prepared figures and/or tables, approved the final draft.
- Brett Favaro conceived and designed the experiments, performed the experiments, authored or reviewed drafts of the paper, approved the final draft.
- Kyla M. Jeffrey performed the experiments, authored or reviewed drafts of the paper, approved the final draft.
- Luis Malpica-Cruz performed the experiments, approved the final draft.

- Natalie Maslowski performed the experiments, approved the final draft.
- Jessica A. Schultz performed the experiments, approved the final draft.
- Nicola S. Smith performed the experiments, analysed the data, authored or reviewed drafts of the paper, approved the final draft.
- Isabelle M. Côté conceived and designed the experiments, performed the experiments, authored or reviewed drafts of the paper, approved the final draft.

## Data Availability

The raw data is available in the Supplemental Files.

## Supplemental Information

Supplemental information for this article can be found online at http://dx.doi.org/10.7717/peerj.6395#supplemental-information.

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
