# Peer review of "Shifting headlines? Size trends of newsworthy fishes"

_PeerJ, doi:10.7717/peerj.6395_

## Round 0.1 · original submission · Major Revisions

· Academic Editor

Major Revisions

As you can see from the reviews, while all three reviewers liked the concept behind your manuscript, their recommendations were quite variable. Two reviewers (1,3) recommend minor revisions while the other reviewer recommendeds rejection. After reading through all three reviews it's my recommendation that you spend some time considerably revising this manuscript.

Reviewer 2 raises some valid concerns about the underlying quality of the data used in the analysis. I agree, and this may or may not be compatible with a revision. If you can find a reasonable means of assessing the quality of the underlying data then you will have addressed to a large extent, this concern. If the reviewer is correct, and the data are simply too unreliable upon which to base an analysis, then the paper probably cannot be published here. Reviewer 3s comments regarding the absence of supporting data to back up your conclusions may perhaps point to the underlying unreliability of the data upon which the analysis is performed.

Both of the other reviewers raise issues associated with your interpretation of the data, particularly as it applies to the charismatic megafishes. Reviewer 1 also was unable to reproduce your figures using the data you provided.

Please read through the comments of all three reviewers carefully. I look forward to reading a substantially-modified version of this manuscript that addresses the concerns raised in this review.

·

Basic reporting

No comment

Experimental design

No comment

Validity of the findings

No comment

Additional comments

Overall I love this paper. The methodology is sound and the paper is overall well written. I have read over the paper several times now looking for any comment or suggestions to make but have nothing to offer. I say this so that this review is not seen a light review but rather the paper is sound. I offer the following as thought and discussion and maybe worth of inclusion?

1. Do you think the lack of pattern reflects low sample sizes and the fact that even in subsets that several species are grouped together? Perhaps this blur any intraspecific patterns?
2. The shifting baseline requires two criteria. The first one describes an overall decline in average size a shift in the distribution. The second criteria is focused on extremes, i.e. the rare event. Although the overall shift in the size distribution may be occurring it does not mean that these already rare events, both the existence of a record holder and its capture, will not happen. You need simply a single individual to escape this pressure. Does the fact the two criteria focus on different aspects of the size spectrum suggest they many not align?
3. It does appear in some of the graphs that the frequency of smaller individuals being reported is increasing even if mean reported size is not. I would suggest potentially running a quantile regression to see if these patterns could be recovered in the lower quantiles. Perhaps this is a draw of the recent as social media and internet have allowed us to report more.

***I had gotten excited about doing this and was going to attach r code and the figure to show you how many of your analyses would likely have significant regressions in the lower quantiles (<25%). However, when I downloaded the data, I was unable to replicate Figure 1. According the the text relative size is reported length divided by Fishbase max length (reported.length.cm / fb.max.tl.cm). This needs to be clarified in the manuscript.

Reviewer 2 ·

Basic reporting

very good

Experimental design

good

Validity of the findings

dubious

Additional comments

This paper is extremely well written and nearly flawless in its presentation. However, the authors used excellent methods and sophisticated statistics and obviously did an exhaustive search of non-scientific literature to analyze questionable, unreliable, dubious, and perhaps incomplete (only literature in English) data. Although the goal of this manuscript is laudable and very important, a manuscript resulting from good methods applied to unreliable data should not be published in a scientific journal. Perhaps some other venue would be more appropriate.

Annotated reviews are not available for download in order to protect the identity of reviewers who chose to remain anonymous.

·

Basic reporting

I don’t have any concerns with the basic reporting categories

Experimental design

As a retrospective study, there’s no experimental design per se. The research question is a worthy one: to test for the presence of a shifting baseline in media reports of size in marine gamefish and “charismatic megafish”.

Validity of the findings

Here I do have some concerns with the conclusions, particularly with respect to the charismatic megafish, rather than the gamefishes.
On line 200, the authors report that relative length of gamefish did not change over time, but that mass increased significantly. This is an odd finding since length and mass are so tightly correlated. Similarly, but in reverse, the relative length of charismatic megafish declined significantly over time, but their relative mass did not. No explanation is given for these unusual results, but they really leap out at the reader so it seems some speculation at least is warranted.
My greater concern is in the interpretation of the results relative to the charismatic megafishes and, specifically, the generalization of well-established reductions in fish length and mass over time among gamefish (refs on line 270) to apply to the megafaunal species for which no such reductions are yet in evidence. I can only speak out of experience with the megafaunal species, but I’m only aware of one published instance that documents decline in size of whale sharks, and that’s the Bradshaw and Holmberg papers cited on line 329-330, which document decrease in length of whale sharks at one site in Australia over time. I don’t think there’s any other whale shark site we’ve been following long enough to detect a change in size over time and I’m not aware of any similar reports for molids, basking sharks or manta rays, so I don’t think the authors can substantiate a pattern of scientifically proven decline in size among megafaunal species as they can for the gamefishes, and thus interpreting changes in their representation in media is tricky.
There’s a second problem with interpretation of the results concerns taxonomic confounding among the megafishes, specifically manta rays and Mola spp.. Up until recently, both mantas and sunfishes were considered monospecific, but we now know that there are probably three manta ray species and at least four Mola spp.. In both cases, the body size varies considerably between species, so how do we interpret historical reports of body size of a “manta ray” or “sunfish” without knowing which species we’re talking about? This does not apply as much to the gamefishes, who have been well studied and are on the whole more easily identified.
Third, something bugs me about the difference between the way gamefishes and megafaunal species come to be represented in media. Gamefishes are deliberately fished, which is an activity that explicitly prizes and seeks to catch the largest individuals. By contrast, the megafaunal species tend only be reported when they are stranded, entangled or caught as bycatch, all methods that are essential neutral to body size, at least relative to targeted gamefishing. How this difference might be expressed in the way the species are recorded in media isn’t clear to me, but it seems like something that should be pointed out.
Finally, in the analysis of changing size relative to IUCN RedList status, the authors chose to compare to the current status of each species, but populations of these species might have been considerably different at the times when media reports were made. Even accounting for this, it occurs to me that the RedList status and relative size ought to be negatively correlated because, as the population shrinks, both tails of the distribution move towards the central tendency, reducing the ratio of average size relative to the maximum. I don’t know how else you would investigate relationship between size and conservation status, but it seems an inevitable consequence of reducing population size that relative size would decline, even if average size does not.

Additional comments

None

---

## Round 0.2 · Minor Revisions

· Academic Editor

Minor Revisions

Reviewer 2 made specific recommendations in the latest review regarding the taxonomy of the ocean sunfishes and mantas. These should be a quick and easy fix. Looking forward to seeing the revision.

·

Basic reporting

I am happy with the revisions and feel the authors have adequately addressed the valid concerns of all three reviewers.

Experimental design

I am happy with the revisions and feel the authors have adequately addressed the valid concerns of all three reviewers.

Validity of the findings

I am happy with the revisions and feel the authors have adequately addressed the valid concerns of all three reviewers.

Additional comments

I think the authors have done a fine job of responding to the authors and I commend them on the additional efforts and analyses.

·

Basic reporting

no additional comments

Experimental design

no additional comments

Validity of the findings

no additional comments

Additional comments

I'm satisfied with the responses to my concerns except the one about taxonomic uncertainty. While the sensitivity analyses are great, the statements about taxonomy of Mola and manta rays are inaccurate, probably due to an over-reliance on Fishbase rather than the taxonomic literature around these two megafaunal groups. Fix that, which is pretty easy, and I'm happy to see this published. PLEASE SEE MY COMMENTS IN THE REBUTTAL FILE IN BLUE

---

## Round 0.3 · accepted · Accept

· Academic Editor

Accept

Thanks for your efforts in bringing this manuscript to publication. I'm happy to say it has now been accepted. Best wishes for 2019 and your future research.

#